# Skin Manifestation of SARS-CoV-2: The Italian Experience

**DOI:** 10.3390/jcm10081566

**Published:** 2021-04-08

**Authors:** Gerardo Cazzato, Caterina Foti, Anna Colagrande, Antonietta Cimmino, Sara Scarcella, Gerolamo Cicco, Sara Sablone, Francesca Arezzo, Paolo Romita, Teresa Lettini, Leonardo Resta, Giuseppe Ingravallo

**Affiliations:** 1Section of Pathology, University of Bari ‘Aldo Moro’, 70121 Bari, Italy; gerycazzato@hotmail.it (G.C.); annacolagrande@gmail.com (A.C.); micasucci@inwind.it (A.C.); vincenza.scarcella@policlinico.ba.it (S.S.); jerrycicco.1@gmail.com (G.C.); lettinit@yahoo.com (T.L.); leonardo.resta@uniba.it (L.R.); 2Section of Dermatology and Venereology, University of Bari ‘Aldo Moro’, 70121 Bari, Italy; caterina.foti@uniba.it (C.F.); romitapaolo@gmail.com (P.R.); 3Section of Forensic Medicine, University of Bari ‘Aldo Moro’, 70121 Bari, Italy; sarasabloneml@gmail.com; 4Section of Gynecologic and Obstetrics Clinic, University of Bari ‘Aldo Moro’, 70121 Bari, Italy; francesca.arezzo@uniba.it

**Keywords:** SARS-CoV-2, skin, COVID-19, WHO, Italian, manifestation

## Abstract

At the end of December 2019, a new coronavirus denominated Severe Acute Respiratory Syndrome Coronavirus 2 (SARS-CoV-2) was identified in Wuhan, Hubei province, China. Less than three months later, the World Health Organization (WHO) declared coronavirus disease-19 (COVID-19) to be a global pandemic. Growing numbers of clinical, histopathological, and molecular findings were subsequently reported, among which a particular interest in skin manifestations during the course of the disease was evinced. Today, about one year after the development of the first major infectious foci in Italy, various large case series of patients with COVID-19-related skin manifestations have focused on skin specimens. However, few are supported by histopathological, immunohistochemical, and polymerase chain reaction (PCR) data on skin specimens. Here, we present nine cases of COVID-positive patients, confirmed by histological, immunophenotypical, and PCR findings, who underwent skin biopsy. A review of the literature in Italian cases with COVID-related skin manifestations is then provided.

## 1. Introduction

At the end of December 2019, a new viral strain of Coronavirus (named SARS-CoV-2) was reported, initially in Wuhan, in the province of Hubei, China [1]. The high infection, low virulence, and asymptomatic transmission provoked its rapid spread beyond the Chinese borders and already by 11 March 2020, a world pandemic [2] was declared by the World Health Organization [2,3]. Up to the end of this study, in February 2021, 105,829,290 confirmed cases had accumulated in the world, with 2,310,605 deaths [4]. In Italy, to date, 2.63 million cases have been confirmed, with 91,003 deaths [5].

SARS-CoV-2 is a virus enclosed in an envelope composed of a single positive pole RNA filament belonging to the Coronavirus family [1]. The virus enters the cells through the angiotensin-converting enzyme 2 (ACE2) receptor on the cell surface [1], and the main site of infection of COVID-19 is the lungs. Patients can present symptoms ranging from mild flu-like complaints up to fulminant pneumonia and potentially fatal respiratory distress (acute respiratory distress syndrome (ARDS)) [6].

The infection incubation time can last up to 14 days after the presumed contact. Typical clinical symptoms include fever, dry cough, sore throat, fatigue, diarrhea, conjunctivitis, hyposmia, and ageusia. The diagnosis is based on clinical history, contact with COVID-19 patients, and clinical symptoms [7]. Confirmation is obtained by performing reverse transcription-polymerase chain reaction (RT-PCR) of the viral RNA on a nasopharyngeal swab or bronchoalveolar fluid. Factors that correlate with a worse outcome are age > 65 years, male sex, cardiovascular disorders, diabetes mellitus, and obesity [8,9,10].

### COVID-19 and Skin

Since the first months of the pandemic, early studies, prevalently Chinese, reported a certain frequency of skin manifestations, albeit in low percentages, in COVID-19 patients. In one work [11], among the 1099 confirmed cases in Wuhan, 0.2% of patients had skin symptoms. The first Italian report was from Lombardy [12] in which among 88 COVID-positive patients, 18 (20.4%) had skin manifestations. In subsequent months, growing numbers of descriptions demonstrated a potential involvement of the skin in COVID-19, and reports in the literature multiplied. Unlike the first reports, in which the skin lesions observed in COVID-19-positive patients were not supported by skin biopsy, an increasing number of biopsies was later performed, leading to a more analytical classification of the various types of skin involvement based on histopathological as well as clinical criteria.

## 2. Materials and Methods

From the beginning of March 2020 until the end of January 2021, we treated 9 patients at the Dermatological Clinic of the University of Bari for skin lesions likely related to SARS-CoV-2. All patients underwent nasopharyngeal throat swabs subjected to RT-PCR that documented the infection. Among these, 3 patients developed a rash similar to erythema multiforme and underwent skin biopsy. A further 3 patients had acro-localized lesions in concomitance with other clinical symptoms of COVID-19. The last 3 patients had diffuse maculopapular lesions. The biopsy specimens obtained were fixed in formaldehyde buffered at 20% and sent to the U.O.C. of Pathology. After appropriate sampling, processing, inclusion in paraffin, and microtome cutting, histological sections with a thickness of about 5 microns were obtained for hematoxylin and eosin (H&E) staining. In addition, 5 other sections were incubated with SARS-CoV-2 spike protein S1 antibody (MA5-36247) in immunohistochemistry (IHC-P), rabbit monoclonal, isotype: IgG, at a concentration of 0.2 µg/mL, with a heat-mediated antigen revelation with citrate buffer at pH 6. All biopsies underwent RT-PCR and detected SARS-CoV-2 with a low copy number.

We then conducted a review of the literature on Italian cases with skin manifestations referring to the electronic databases PubMed and Web of Science from the start of the pandemic until the end of December 2020, using the terms “COVID-19” or “nCov-19” in combination with “skin” or “cutaneous manifestations”, or “eruption”, “rash”, “exanthem”, “urticarial”, “chilblain”, “livedo”, and “purpura” with “Italy”, or “Italian”, in order to collect the reports of skin manifestations described in patients with COVID-19 in Italy. A summary table is provided in Appendix A.

## 3. Results

The three patients with erythema-multiforme-like lesions were aged between 23 and 44 years and had mild flu-like symptoms together with the rapid development of skin lesions, featuring erythematous edematous papules localized firstly on the face and palms of the hands, which progressively changed to erythematous purplish patches with a brownish center; finally, the typical target lesions developed and rapidly spread to the entire skin surface (Figure 1a–c). The skin biopsies showed a perivascular lichenoid lymphocytic infiltrate, prevalently localized in the superficial and middle derma, rarely involving the basal layer, and not completely obscuring the dermo–epidermal junction (Figure 1d). Few, rare eosinophils were present. Immunostaining with anti-SARS-CoV-2 antibodies showed positivity for the viral spike proteins at the level of the vascular endothelium, with a characteristic granular cytoplasmic positivity (Figure 1e). We performed a SARS-CoV-2 PCR in skin biopsies with positive results.

Three patients between the ages of 17 and 32 years had developed painless erythematous-crusted, acro-localized plaques (Figure 2a), which were quite similar in all cases. They were almost entirely symptom-free and complained only of a slight headache. Histological patterns were quite different from those of our previous observations, showing predominant edema of the middle and deep dermis, accentuated elastolysis (confirmed by histochemical staining for orcein, not shown) and fragmentation of the collagen fibers that seemed to form brownish detritus (Figure 2b,c). There was also accentuated thrombosis of the vessels of the superficial dermal capillary plexus, with a hobnail-type endothelium (Figure 2d) and moderate chronic inflammatory infiltrate, prevalently consisting of lymphocytes, often involving the vessel wall and the eccrine sweat glands. The subcutaneous layer was only faintly affected. Immunohistochemical investigations were also positive for SARS-CoV-2 in this case in the vascular endothelium and eccrine sweat glands (Figure 2e). Immunohistochemistry was positive for SARS-CoV-2 PCR in skin biopsies too at the level of the vascular endothelium and eccrine sweat glands (Figure 2e). Cutaneous specimens underwent molecular tests by RT-PCR, which produced positive results for SARS-CoV-2 in all three cases.

The last three patients had maculopapular lesions, with no clinical or histological data suggesting a specific dermatosis. However, despite the lack of specific characteristics, immunohistochemistry and PCR conducted on the skin biopsies obtained from these patients showed a strong positivity for viral spike proteins, further confirming the presence of the virus in the skin (Figure 3).

## 4. Discussion

The early studies conducted in China reported low frequencies of skin manifestations in the course of COVID-19 infection: Guan et al. were the first authors to describe, among 1099 confirmed cases, the involvement of the skin during SARS-CoV-2 infection in 0.02% of cases [11]. In Italy, Recalcati et al. described skin signs in 18/88 patients (20.4%) positive got the COVID-19 test: 8 at disease onset and 10 after hospitalization [12], although the very first authors were Mazzotta et al., who described acro-localized lesions in young adolescents in mid-March 2020 [13]. At the end of April 2020, a letter from Thailand [14] reported a high rate of skin involvement in patients with SARS-CoV-2. Lei et al. correlated the low incidence of skin manifestations to a more mild form of the disease in the Tibetan highlands [15]. In April 2020, Gianotti et al. reported the first three Italian cases of skin manifestations of COVID-19 supported by histological findings in the course of COVID-19 infection. They described maculopapular eruptions, sometimes accompanied by pruritus, affecting the upper and lower limbs as well as the trunk [16,17]. In May 2020, Gianotti et al. [18,19] described a further five patients with various skin signs, including exanthematous papular eruptions (three patients) and a maculopapular skin rash (two patients) localized on the trunk and arms. Further, in May 2020, Tosti et al. [20] described four cases of erythematous chilblain-like nodules on the heels and toes. These data were corroborated by Colonna et al. who described four cases of children with erythematous macules on their feet [21]. Then, Genovese et al. described the case of a child with a generalized maculopapular rash, accompanied by itching, of the morbilliform type [22]. At the end of May, Hachem et al. [23] reported their experience of 19 adolescents with chilblain-like lesions on their toes. In June 2020, Marzano et al. [24] reported a multicentric experience of 22 cases of patients with diffuse papulovesicular lesions on the trunk. Diotallevi et al. [25], also in June, presented three cases of patients aged between 12 and 64 years showing erythematous urticarial macules on the trunk (two adult patients) and chilblain-like lesions in the feet of an 12 year old adolescent.

In July 2020, Tammaro et al. [26] described two COVID-19-positive patients with herpetiform lesions on the trunk. Tammaro et al., also in July 2020, described a case of acro-localized necrotic lesions in a patient with fatal COVID-19 pneumonia [27]. In the same month, Piccolo et al. described the preliminary results of 63 patients with chilblain-like lesions [28].

The reports in August 2020 by Gianotti et al. and Recalcati et al. described the interesting cases [29,30] of eight patients with a positive diagnostic test (RT-PCR), who developed various different lesions, namely maculopapular exanthema, varicelliform-like rash and maculopapular lesions aggregated into plaques. A second group of patients (n = 14), mainly adolescents and young adults, developed chilblain-like lesions together with erythema pernio and multiforme-like lesions. Although all these patients had a negative molecular test, they presented these skin manifestations in concomitance with the massive spread of the COVID-19 pandemic occurring at that time. A last group of patients in the same study reported subjects who had been unable to undergo the SARS-CoV-2 molecular testing for various and disparate reasons, but had also developed maculopapular lesions (in particular, in two HIV-positive subjects), sometimes aggregated into plaques. De Giorgi et al. described 17 cases of urticarial rash in 17 swab-positive patients for SARS-CoV-2. [31]. In the same month, Castelnovo et al. described symmetric cutaneous vasculitis in COVID-19 pneumonia [32]. In August and September, further chilblain-like lesions were described in young patients by Locatelli et al. [33], Colonna et al. [34], and Maniaci et al [35]. Freeman et al. recorded in their work [36] a case of an adult patient with erythema pernio from Italy, along with numerous other cases from seven other countries.

In September–October 2020, Brazzelli et al. [37] described eight COVID-19-positive patients who, after having a negative nasopharyngeal swab (NPS), developed a macular exanthem with a distinct pattern. In September 2020, Guarneri et al. described 13 patients with urticarial eruptions (two patients), panniculitis (three patients), erythematous rash (two patients), and chilblain lesion (one patient) [38].

In October 2020, Gaspari et al. [39] reported their experience of 18 patients with signs such as an exanthematous rash on the body (nine patients), acral vasculitis-type eruptions in six adolescents, two cases of polymorphous erythema-like urticaria, and one case of varicelliform rash. In the same month, Di Nunno et al. [40] reported their experience in a COVID hub in which 34 cases of COVID-related skin manifestations had occurred: the most recurring was skin dryness (15 patients, 16.0%), irritant contact dermatitis (5 patients, 5.2%), seborrheic dermatitis (4 patients, 4.2%), morbilliform rashes (4 patients, 4.2%), petechial rashes (3 patients, 3.1%), and widespread hives (3 patients, 3.1%). In any case, the authors recognized the possibility that some of these skin manifestations can be traced to therapies or situations other than SARS-CoV-2 infection. At the end of October 2020, Caputo et al. described an interesting case of generalized purpuric eruption with histopathologic features of leukocytoclastic vasculitis in a patient severely ill with COVID-19 [41].

Since November 2020, new reports of skin manifestations have started to appear in subjects who contracted COVID-19 but were negative for SARS-CoV-2 at the time. Balestri et al. [42] reported one case of true acral necrosis in one patient, suggesting possible immunological mechanisms at the basis of the cytochemical cascade responsible, amongst others, for at least part of the skin manifestations. Rossi et al. [43] described a case of an acute, disseminated skin rash in a young patient testing positive for SARS-CoV-2. Finally, Negrini et al. [44] reported, at the end of November 2020, a case of bullous hemorrhagic vasculitis concomitant with COVID-19 infection. Carugno et al. hypothesized an involvement of interleukin 17 in the genesis of erythematoedematous morbilliform rash in a patient previously suffering from psoriasis and treated with the monoclonal antibody secukinumab [45]. Interestingly, in the same month, Proietti et al. described for the first time a probable auricular perniosis in a 35 year old patient [46]. Annunziata et al. reported four cases of asymptomatic erythematous pomphoid skin rash on trunk and limbs [47]. Finally, Promenzio et al. reported four cases of erythema pernio-like lesions in SARS-CoV-2 positive adolescents [48].

In December 2020, Quaglino et al. described a case of different skin manifestations present in the same patient over different time periods, suggesting two different pathogenetic hypotheses potentially responsible [49]. However, the possible correlation between chilblain-like skin lesions and COVID-19 is still a matter of debate, as evidenced by the work of Denina et al. [50].

In January 2021, Montanari et al. described a case of erythema annulare centrifugum with anosmia and ageusia in a SARS-CoV-2-exposed patient successfully treated with doxycycline [51]. In the same month, Pezzarossa et al. described how the viral infection could unmask manifestations of acute generalized exanthematous pustulosis (AGEP) in patients treated for COVID-19 pneumonia [52].

In February 2021, Patrì et al. reported two cases of rashes (purpuric and suberythrodermal) in two COVID-positive patients, postulating an involvement of endothelial dysfunction as a possible pathogenesis mechanism of COVID-related skin manifestations [53].

Recently, Rossi et al. [54] postulated a possible common mechanism to explain a relative increase in alopecia areata cases in a COVID-positive patient. Currently, the research direction is beginning to describe the onset of a red flag in chilblain of previously infected individuals [55].

The situation is still changing rapidly [56,57,58,59,60] but a growing knowledge of the etiopathogenetic mechanisms potentially responsible for these skin manifestations is gradually emerging. Below, we summarize the clinical and histopathologic characteristics described to date.

### 4.1. Chilblain-Like Lesions on Fingers and Toes

Edematous and erythematous eruptions similar to chilblains have been observed in milder cases of COVID-19, particularly in young children and young adults, which tend to disappear without scarring at the end of the infection [48]. Chilblain-like eruptions are mostly asymmetrically distributed, sometimes associated with itching or pain (in 22% and 11% of cases, respectively) [48], and the mean time of onset of the symptoms is after about 10 days. Histopathology of the lesions shows mainly epidermal basal layer vacuolation, papillary dermis edema and erythrocyte extravasation, a perivascular and peri-eccrine dermal lymphocytic infiltrate, and mucin deposits in the dermis and hypodermis. Dermal vessel thrombi are sometimes present [23].

### 4.2. Acro-Ischemic Lesions

Patients affected with more severe COVID-19 symptoms showed a hypercoagulation state and disseminated intravascular coagulation, with laboratory tests revealing increased levels of D-dimer, fibrinogen and fibrinogen degradation products, and a prolonged prothrombin time. These critical patients presented acro-ischemia with cyanosis of the fingers and toes, skin blisters, and sicca gangrene. [49]. Histology of skin biopsies showed degeneration of the dermal collagen with elastolysis and abundant mucin deposition, and phenomena of the dermal collagen with elastolysis and abundant mucin deposits. Dermal edema and the presence of microthrombi in various stages of organization were constant findings in the dermal vessels.

### 4.3. Rash with Petechiae and Purpuric Rash

Patients with a morbilliform rash were mainly affected in the buttocks, popliteal pits, proximal anterior thighs and lower abdomen, more rarely in the crural folds, face, palmoplantar skin and mucosa [47]. Skin biopsies in these cases revealed a superficial perivascular lymphocytic infiltrate with abundant extravasation of red cells and focal papillary edema; the epidermis showed focal parakeratosis and dyskeratotic cells. There were no signs of thrombotic vasculopathy [46].

### 4.4. Chickenpox-Like Rash

This type of skin manifestation recurred in a limited number of cases [42] and consisted of varicella-like rashes with small monomorphic blisters located mainly on the trunk, particularly present in middle-aged COVID-19 positive men [12].

### 4.5. Urticarial and Erythema Multiforme-Like Rash

Various patients presented an urticarial rash that affected various skin locations and was sometimes accompanied by pyrexia, so much so that it was considered indicative of COVID-19 disease. Patients tended to present these lesions early in the infection [48]. In contrast, erythema multiforme-like lesions have been shown to be more common in children and associated with a milder course of COVID-19. From the histological point of view, the descriptions in the literature [16,17,48] do not differ from the more traditional descriptions of such cutaneous manifestations.

### 4.6. Maculopapular Rash

This skin manifestation is most frequent in the literature in Italy, mostly affecting adults, associated with a severe COVID-19 disease course. In the cases analyzed, the maculopapular rash could affect any body district, often becoming widespread [12,16,17,18,19,23,27]. More rarely it affected children [42] and had a mean time to onset of 9 days [42]. The histopathological pictures proved very varied [42,43], but common elements were represented by superficial perivascular lymphocytic infiltrate in combination with dilated vessels in the upper and middle dermis.

## 5. Conclusions

New and increasingly detailed descriptions of skin lesions are continually published in literature, and this makes the study of skin involvement during infection more intriguing, but also more complex. One of the most debated aspects concerns the pathogenetic mechanism of COVID-related lesions and whether this is due to a direct effect of the virus rather than being secondary to the immune response related to the infection [44]. New studies, research, and works will be necessary to be able to unravel the complex and intricate mechanism of damage to the skin, to propose a simple and useful clinicopathological classification, and to find possible therapeutic paths.

## Figures and Tables

**Figure 1 jcm-10-01566-f001:**
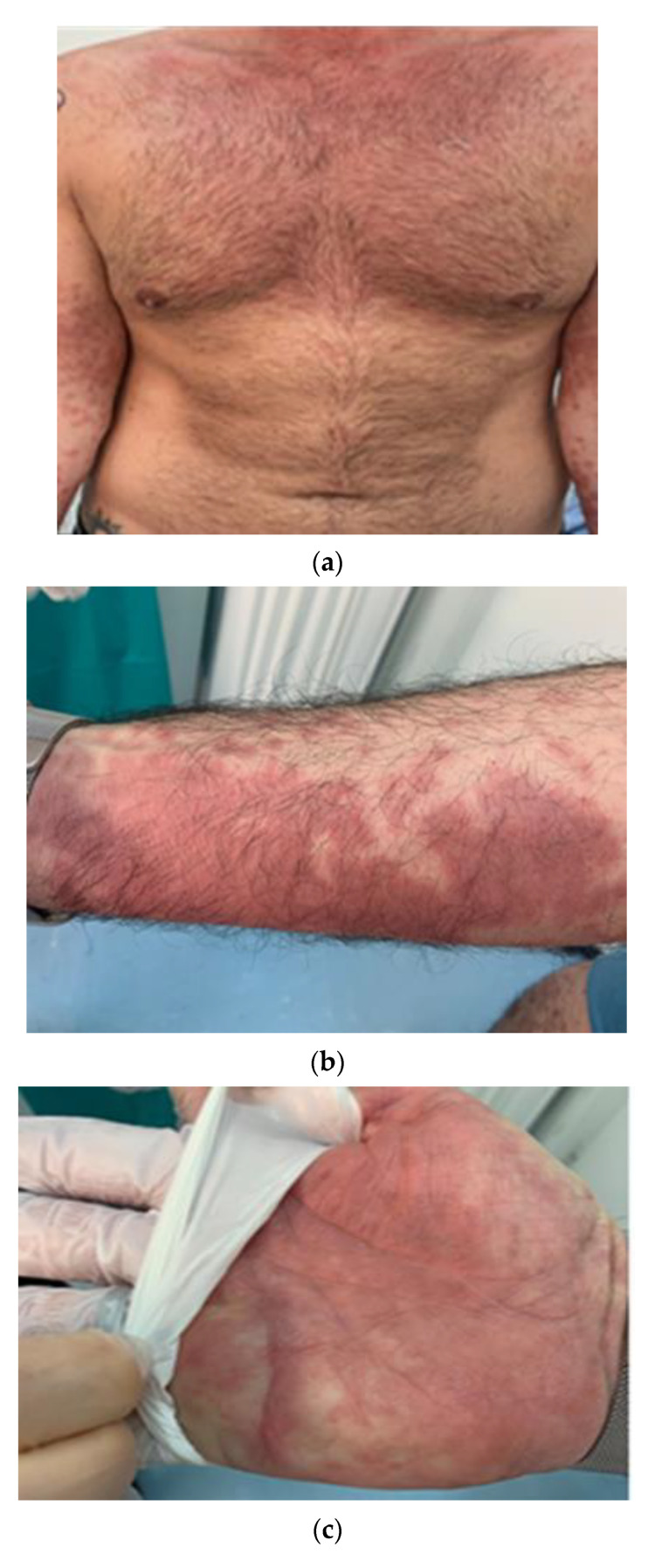
(**a**) Example of a skin rash with polymorphic erythema characteristics affecting the chest, arms and hands. Erythematous patches with irregular, polycyclic contours. (**b**) Detail of the erythema on the forearm. (**c**) Detail of the erythema on the hand. (**d**) Histopathological findings of erythema-multiforme-like lesions. Note the perivascular lymphocytic infiltrate and focal dermal mucinosis (hematoxylin and eosin (H&E), original magnification 100×). I Details of a SARS-CoV-2 spike protein in endothelial cells from erythema-multiforme-like lesions (immunostaining for SARS-CoV-2 spike proteins, original magnification 400×).

**Figure 2 jcm-10-01566-f002:**
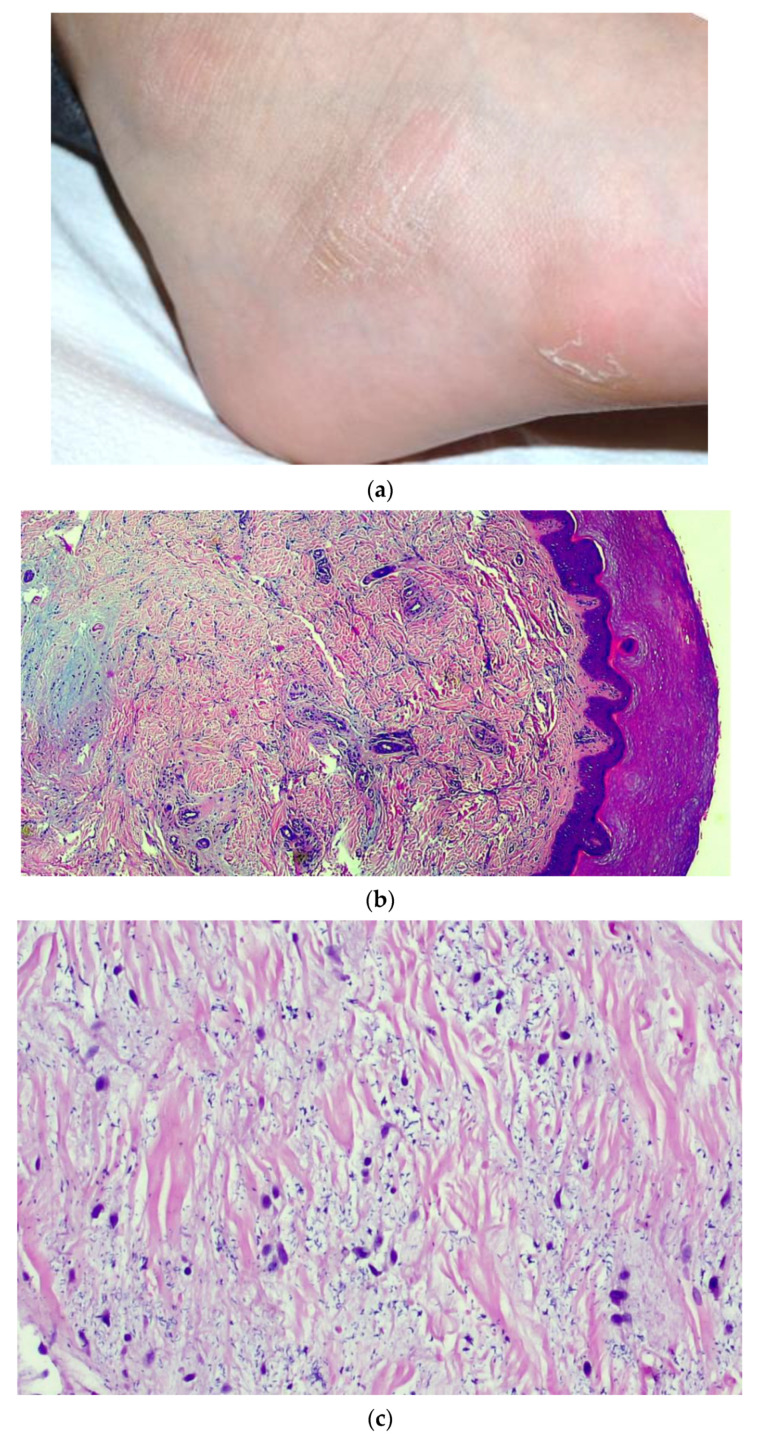
(**a**) Example of erythematous nodules that were hard, elastic, and painless on palpation, present on the lateral surface of the feet. (**b**) Histopathological overview of acral lesions (H&E, original magnification, 100×). (**c**) Details of fragmentation and final dissolution of the collagen and elastic fibers, with the formation of blackish detritus. Presence of reactive fibroblasts. (**d**) Detail of blood vessels with a prominent endothelium of hobnail type. (H&E, original magnification 400×). (**e**) Positive immunostaining in eccrine sweat glands (original magnification, 200×).

**Figure 3 jcm-10-01566-f003:**
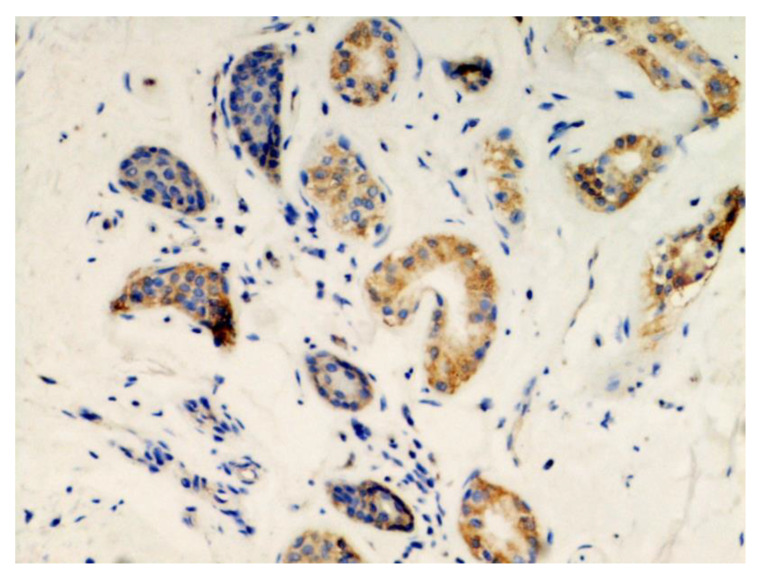
Immunohistochemical details of patients with COVID-19-linked maculopapular eruptions (original magnification, 400×).

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
