# Peer review of "Skin Manifestation of SARS-CoV-2: The Italian Experience"

_jcm, 2021, doi:10.3390/jcm10081566_

Round 1
Reviewer 1 Report
The authors present their well-documented case series of patients with SARS-CoV-2 infection with cutaneous manifestations. The patients were investigated with skin biopsies, immunohistochemical investigations and PCR.
The intention to create a review of the 2020 Italian case series is very interesting. However, the bibliographical research has to be improved. For example, I noticed that a paper by the group of Bergamo was included; it reports the incidence of infection in patients suffering from bullous diseases, not the bullous manifestations of SARS-CoV-2 (https://doi.org/10.1111/bjd.19266) and instead the cases of skin manifestations are missing (https://doi.org/10.1111/dth.14011, https://doi.org/10.1111/jdv.16613).
Also missing are these case reports from the Pavia group (https://www.jle.com/10.1684/ejd.2020.3855) or these other reports (https://doi.org/10.23736/s0392-0488.20.06735-8, https://doi.org/10.1111/ijd.15356, https://doi.org/10.1684/ejd.2020.3925, ...).
I therefore recommend completing the literature review, which could be very interesting.
Author Response
Dear Reviewer,
We appreciate your helpful comment.
Best Regards
Giuseppe Ingravallo
On behalf of the co-authors
Review n’ 1: “The intention to create a review of the 2020 Italian case series is very interesting. However, the bibliographical research has to be improved. For example, I noticed that a paper by the group of Bergamo was included; it reports the incidence of infection in patients suffering from bullous diseases, not the bullous manifestations of SARS-CoV-2 (https://doi.org/10.1111/bjd.19266) and instead the cases of skin manifestations are missing (https://doi.org/10.1111/dth.14011, https://doi.org/10.1111/jdv.16613).
Also missing are these case reports from the Pavia group (https://www.jle.com/10.1684/ejd.2020.3855) or these other reports (https://doi.org/10.23736/s0392-0488.20.06735-8, https://doi.org/10.1111/ijd.15356, https://doi.org/10.1684/ejd.2020.3925, ...)”.
Answer 1: Dear reviewer, thank you very much first of all for your invaluable comments that we have followed in a slavish way to make our manuscript better. We have rectified the items that had been misquoted and added all the items suggested in both the Discussion, the Bibliography and the Table. We hope we have followed his valuable advice correctly. Thank you very much, the authors.
Reviewer 2 Report
New studies, research, and works will be necessary to be able to unravel the complex and intricate mechanism of damage to the skin, to propose a simple and useful clinicopathological classification, and to find possible therapeutic paths. Very interesting and practical work in this special pandemic time! Congratulations!
Author Response
Dear Reviewer,
We appreciate your helpful comment.
Best Regards
Giuseppe Ingravallo
On behalf of the co-authors
Review n’2: “New studies, research, and works will be necessary to be able to unravel the complex and intricate mechanism of damage to the skin, to propose a simple and useful clinicopathological classification, and to find possible therapeutic paths. Very interesting and practical work in this special pandemic time! Congratulations!”.
Answer 2: Dear reviewer, we are thrilled by your words. We tried our best. Thanks so much!
Reviewer 3 Report
Signs and symptoms of the new viral infection might range from an absence of symptoms to severe and sometimes, life-threatening condition.
One of these systemic symptoms is dermatological manifestations. Some patients with SARS-CoV-2 were observed to have some cutaneous symptoms such as urticaria spreading over the body, erythematous rash, skin vesicles, similar to chickenpox infection. These dermatological symptoms were commonly reported all over the body, particularly over the trunk. Also, patients with SARS-CoV-2 complained of itching of varying severity.
The manuscript is interesting, novel, clear and the English language requires a moderate editing.
However, I have these following major concerns:
- A major limitation is that the study sample is not fully representative of the general population, as it represents a self‐selected group of individuals. The study sample is little and also composed predominantly of white individuals compared with those observed in hospital settings. Moreover, the data are collected only during the milder phases of the disease and in individuals who did not require hospitalization as, when health deteriorates, logging often stops.
- A second limitation, the authors did not consider rare dermatological presentations. The main aim of this study is not to provide an exhaustive description of SARS‐CoV‐2 cutaneous manifestations but to raise awareness of the high prevalence of common COVID‐19 rashes, which can sometimes appear earlier than other COVID‐19 symptoms or be the only symptom.
- Thirdly, some of these cutaneous manifestations could have been caused by adverse reactions to drugs used to treat SARS‐CoV‐2 and/or for other purposes.Are there any pre‐existing skin diseases of the patients under investigation to consider if they influenced the rate of the reported rashes?
- Any therapeutic options that could help in managing the hyperactivity of the immune system in COVID-19? For example: low-dose systemic corticosteroids, combined with nonsedating antihistamines.
- I recommend including a list of all abbreviations used in the text and paying attention to write the full names of the acronyms reported in the text.
- I recommend including also a table with all the patient specifications.
- the authors should cite the following review (Palma G, Pasqua T, Silvestri G, et al. PI3Kδ Inhibition as a Potential Therapeutic Target in COVID-19. Front Immunol.2020;11:2094. Published2020Aug21 (doi:10.3389/fimmu.2020.0209).
Author Response
Dear Reviewer,
We appreciate your helpful comments; we believe that our manuscript has been improved greatly through the incorporation of your suggestions.
Best Regards
Giuseppe Ingravallo
On behalf of the co-authors
Review n’3: “The manuscript is interesting, novel, clear and the English language requires a moderate editing”.
Answer 1: Dear reviewer, thank you very much for your beautiful comment. We have proceeded to double-check the English as indicated by you. Thanks so much.
Review n’3: “A major limitation is that the study sample is not fully representative of the general population, as it represents a self‐selected group of individuals. The study sample is little and also composed predominantly of white individuals compared with those observed in hospital settings. Moreover, the data are collected only during the milder phases of the disease and in individuals who did not require hospitalization as, when health deteriorates, logging often stops”.
Answer 2: Dear reviewer, thank you very much for your comment which we fully share. Our case series relates to non-hospitalized patients with mild / moderate manifestations of the disease. We hope to be able to increase the number of cases in a potential and next work. Thanks so much.
Review n’3: “A second limitation, the authors did not consider rare dermatological presentations. The main aim of this study is not to provide an exhaustive description of SARS‐CoV‐2 cutaneous manifestations but to raise awareness of the high prevalence of common COVID‐19 rashes, which can sometimes appear earlier than other COVID‐19 symptoms or be the only symptom”.
Answer 3: Dear Reviewer, I absolutely agree with your observation. We have included in the review of the literature only the cases in which "beyond a reasonable doubt" the skin lesions could be attributed to Covid-19. However, at his suggestion we have added other bibliographical entries of rarer dermatological manifestations. Thanks so much.
Review n’3: “Thirdly, some of these cutaneous manifestations could have been caused by adverse reactions to drugs used to treat SARS‐CoV‐2 and/or for other purposes. Are there any pre‐existing skin diseases of the patients under investigation to consider if they influenced the rate of the reported rashes?”
Answer 4: Dear reviewers, in this case too we believe that your observation can grow our work. In our 9 cases, the patients did not suffer from any recognized pre-existing skin disease. Clinical, histopathological and PCR data gave us greater certainty that these lesions were likely to be related to SARS-CoV-2. In any case, for scientific correctness, in conducting the review of the literature we proceeded to specify when the same authors cited had doubts regarding the correlation between skin lesions and Covid-19. Thanks so much.
Review n’3: Any therapeutic options that could help in managing the hyperactivity of the immune system in COVID-19? For example: low-dose systemic corticosteroids, combined with nonsedating antihistamines.
Answer 5: Correct observation, as many dermatological works are moving in this direction. Thanks again!
Review n’3: “I recommend including a list of all abbreviations used in the text and paying attention to write the full names of the acronyms reported in the text”.
Answer 5: Dear reviewer, we have added a list of abbreviations used in the text and double-checked that all abbreviations are first mentioned in the text. Thanks so much.
Review n’3: I recommend including also a table with all the patient specifications.
Answer 6: Dear reviewer, we have integrated and completed a table with all the specifications relating to the number of patients, type of skin manifestation and topographical location. Thanks so much.
Review n’3: The authors should cite the following review (Palma G, Pasqua T, Silvestri G, et al. PI3Kδ Inhibition as a Potential Therapeutic Target in COVID-19. Front Immunol.2020;11:2094. Published2020Aug21 (doi:10.3389/fimmu.2020.0209).
Answer 7: Done, thank you very much.
Round 2
Reviewer 1 Report
I congratulate the authors on the reviews conducted so far, however the literature review has not been fully expanded as requested, only the 3 articles I indicated as examples have been included.
There are others that can be included following the criteria they indicate in the materials and methods. From a quick search the first ones I found are: https://doi.org/10.1111/dth.14011, https://doi.org/10.1111/jdv.16613, https://doi.org/10.12659/ajcr.925813, https://doi.org/10.1111/ajd.13400, ....
I would ask the authors to make a last effort to complete the search if they find other articles in addition to those indicated.
Author Response
Dear reviewer,
thank you very much for the compliments. We made a final further effort and covered other articles (many of them in the first months of 2021) to complete a correct and complete review of Italian literature. We hope it will meet your favor.
A warm and affectionate greeting,
Giuseppe Ingravallo
On behalf of the co-authors
Reviewer 3 Report
The authors addressed critically to all my concerns.
Author Response
Dear Reviewer,
thank you very much
Best Regards
Giuseppe Ingravallo
On behalf of the co-authors